# How Do Young Women with Cancer Experience Oncofertility Counselling during Cancer Treatment? A Qualitative, Single Centre Study at a Danish Tertiary Hospital

**DOI:** 10.3390/cancers13061355

**Published:** 2021-03-17

**Authors:** Line Bentsen, Helle Pappot, Maiken Hjerming, Lotte B. Colmorn, Kirsten T. Macklon, Signe Hanghøj

**Affiliations:** 1Department of Oncology, Copenhagen University Hospital, Rigshospitalet, Blegdamsvej 9, 2100 Copenhagen, Denmark; Helle.Pappot@regionh.dk; 2Department of Hematology, Copenhagen University Hospital, Rigshospitalet, Blegdamsvej 9, 2100 Copenhagen, Denmark; Maiken.Hjerming@regionh.dk; 3The Fertility Clinic, Copenhagen University Hospital, Rigshospitalet, Blegdamsvej 9, 2100 Copenhagen, Denmark; Lotte.Berdiin.Colmorn@regionh.dk (L.B.C.); Kirsten.Louise.Tryde.Macklon@regionh.dk (K.T.M.); 4Center of Adolescent Medicine, Department of Pediatrics and Adolescent Medicine, Copenhagen University Hospital, Rigshospitalet, Blegdamsvej 9, 2100 Copenhagen, Denmark; Signe.Hanghoej@regionh.dk

**Keywords:** adolescents and young adults (AYAs), cancer, fertility, oncofertility counselling, qualitative study

## Abstract

**Simple Summary:**

Adolescents and young adults (AYAs) diagnosed with cancer undergo a range of cancer treatments with intension of being cured. It is well-documented that cancer treatment induces a risk of infertility due to ovarian damage. In recent years fertility preservation options for female AYAs with cancer have developed with opportunities for pregnancy after completion of cancer treatment. Despite international guidelines, a high level of evidence underpins the insufficiency of fertility counselling towards this patient group. The aim of this qualitative study was to examine the experience of fertility counselling from the AYAs perspective, and we strive to increase the attention towards the need of improvement and developing national guidelines at international level to ensure adequate and uniform fertility information.

**Abstract:**

Background: Adolescents and young adults (AYAs) with cancer are at risk of therapy-induced infertility. The importance of initial and specialized fertility counselling to this patient group is undisputed. Despite international guidelines, oncofertility counselling is still inadequate. The purpose of this study was to examine how female AYA cancer patients and survivors experienced initial and specialized oncofertility counselling, and to present their specific suggestions on how to improve the oncofertility counselling. Methods: Twelve individual semi-structured interviews were performed with AYAs aged 20–35 with cancer or who were survivors. Participants were recruited via a youth support centre and social organization for AYAs with cancer. Data was analysed using thematic analysis. Results: Three main themes were found: Support is needed for navigating the fertility information jungle; The doctor’s approach determines the content of the patient consultation; Inadequate and worrying information causes mistrust and frustration. Conclusion: Results indicate a continuing problem regarding insufficient oncofertility counselling to AYAs with cancer. To ensure adequate and uniform information, especially in the initial oncofertility counselling, national guidelines for oncology specialists are suggested including multidisciplinary effort and collaboration between oncology and fertility specialists in mind. In addition, participants suggest focus on communication skills.

## 1. Introduction

Adolescents and young adults (AYAs) with cancer are considered to be a patient group with physical, emotional and social challenges specific to their age, throughout treatment and in survivorship [1]. A cancer diagnosis is not only potentially life-threatening but, for this patient group, the disease also has an enormous impact on their current life situation and health-related quality of life [2,3]. In recent years, there has been a worldwide increased focus on this patient group’s treatment and support needs, including relationships, fertility and family planning [4].

It is well known that one consequence of cancer treatment, especially alkylating chemotherapy and radiation therapy, is the risk of reduced fertility—for both genders [5,6,7,8]. However, recent decades have seen great development in fertility preservation, e.g., cryopreservation of oocytes, embryos or ovarian tissue, which has therefore increased chances of future fertility and pregnancy [9,10,11,12].

International guidelines recommend that initial, pre-treatment oncofertility counselling should be provided to young, fertile cancer patients as soon as possible upon diagnosis, and health professionals working in this field must be prepared for this conversation [5,10,13,14]. The counselling should include thoughts about future family planning, the potential impact of cancer treatment on fertility, the chances of pregnancy and the possible need for effective contraception after cancer treatment [14]. It has been recommended that patients identified to be at risk of infertility who are interested in fertility preservation be immediately referred to a fertility specialist, and if possible, prior to gonadotoxic treatment for specialized fertility counselling [13,15].

AYAs with cancer and survivors are concerned about their fertility [16,17,18], and they have a strong desire for fertility information from their oncology specialists at diagnosis [19]. However, despite an increased focus on the need for oncofertility counselling, research shows that not everyone is offered proper counselling. A study from the US showed that only 60% of AYAs with cancer received pre-treatment fertility counselling, of which only 13% were referred to further counselling by a fertility specialist [20]. In another study, fewer than half of fertile patients with cancer were informed about fertility preservation, and even fewer received it [9].

Despite international guidelines and AYAs’ need for fertility counselling, initial oncofertility counselling is not always conducted systematically to patients treated in oncological and haematological settings, even though being offered specialized oncofertility counselling is associated with increased quality of life for AYA survivors [21]. New evidence argues substantial variations in the clinical guidelines reducing consistency and timely implementation of effective interventions for fertility preservation across institution [22]. This can be as a result of various physician-related, institution-related and/or patient-related barriers. National legislation and access to funding can also represent a barrier to receiving fertility preservation treatment [10,23].

In Denmark, approximately 900 women a year aged 18–39 are diagnosed with cancer [24]. At the University Hospital of Copenhagen, Rigshospitalet (The Copenhagen University Hospital, Rigshospitalet, is situated in the Capitol of Denmark, being the largest hospitals in the country with approximately 80.000 hospitalizations and 1.2 million ambulant visits every year. Cancer patients are diagnosed and treated in particular at the departments of haematology, paediatrics and adult cancer in close collaboration with surgical and intern medical departments at the hospital [25]), in Denmark, where this study was conducted, the aim is to offer fertility preservation and pre-treatment fertility counselling to young men and women of fertile age, before a potentially sterilizing cancer treatment. However, the services are only offered if the patient is referred to fertility specialists by the oncology or haematology department. An urgent call for systematic initial and referral to specialized oncofertility counselling has been addressed by AYAs with cancer at the hospital. Their concerns are brought up at activities in an AYA cancer supportive facility, Kræftværket (Kræftværket is a youth support centre and social organization situated at the Copenhagen University Hospital, Rigshospitalet. It is developed in 2015, providing care and treatment practices and allowing AYAs with cancer to connect with peers and other AYAs with cancer. Kræftværket also participates in the ongoing development of a multidisciplinary collaboration across adolescent medicine, paediatric and adult cancer departments, to incorporate knowledge sharing, teaching and involvement of relevant health professionals and to coordinate political and administrative activities surrounding AYAs with cancer. A full-time Youth Coordinator and several youth ambassadors are managing events and activities of Kræftværket, as well as serving as an ambassador and supporter of young patients. AYAs with cancer are directly involved in making decisions regarding activities and resources available at Kræftværket. Every three months, a youth panel meeting is held to discuss and receive feedback about Kræftværket with AYA cancer patients and survivors to make changes reflecting the needs of young patients. From 2015–2020 an amount of 7700 visits at Kræftværket is recorded [26].), where youth panel meetings are held [26].

The aim of this study was twofold: (1) to examine how female AYA cancer patients and survivors experienced initial and specialized oncofertility counselling; and (2) to present their specific suggestions on how to improve the oncofertility counselling.

## 2. Materials and Methods

### 2.1. Setting

The study was carried out during September 2020 at ”Kræftværket”, the youth support centre and social organization for AYAs with cancer at the University Hospital of Copenhagen, Rigshospitalet.

### 2.2. Recruitment and Participants

Inclusion criteria were female cancer patients and survivors aged 18–39. Recruitment was facilitated through Kræftværket, via a written invitation posted on a closed Facebook group managed by Kræftværket. The closed Facebook group has approximately 320 active members, all AYAs with cancer or survivors. At the time of recruitment, twelve participants who meet the inclusion criteria signed up to take part in the study. The participants were aged 20–35 (mean age: 28).

### 2.3. Data Collection

The interview guide comprised questions to elicit thoughts about future family planning and pregnancies, expectations of oncofertility counselling, and experiences of the counselling during and after cancer treatment (see Table 1).

Twelve individual semi-structured interviews were performed and lasted, on average, 33 min (between 23 and 59 min). As data saturation [27] was achieved after the first ten interviews, it was decided not to recruit additional participants. All interviews were recorded electronically and transcribed verbatim in Danish by LB. Quotations were subsequently translated into English by a professional translator.

### 2.4. Analysis

The analysis of this study was inspired by a thematic analysis approach rooted in Malterud’s systematic text condensation [28] and Braun and Clark’s reflexive thematic analysis [29]. The study took a phenomenological approach and data was analysed inductively, as coding and themes arose from the content in the data. The following steps were conducted: (1) the recorded interviews were read several times by LB and SH while noting possible codes; (2) relevant codes for the research question were discussed between LB and SH after several readings; (3) preliminary themes were developed by LB and SH on the basis of the codes; (4) LB determined the content of each preliminary theme and chose a title for each theme; (5) the research group discussed the themes and allocation of codes until agreement on the final themes was reached; and finally (6) LB wrote the final content for these themes and, in collaboration with SH, prepared the article, based on the analysis. To validate the design, the interview guide ensured that the participants were asked the same number and variety of questions. Rigour and reliability were ensured using the above mentioned 6-step analysis, with repeated readings of the transcripts. Furthermore, themes were found independently by, respectively, the first and last author, and all themes were discussed in the author group to secure consensus.

### 2.5. Ethics

The project was approved 1 September 2020 by the local Data Protection Agency (P-2020-849). The participants were informed about the aim and design of the study, including information about anonymity and that they could withdraw their consent at any time if they wished without adverse consequences for their treatment. They were also informed that the interviews would be recorded electronically. Written informed consent was obtained from all participants.

### 2.6. Definitions

In this article, doctors at the respective oncology and haematology departments are defined as oncology specialists. Doctors at the fertility clinic are defined as fertility specialists.

Oncofertility counselling is divided into initial counselling and specialized counselling. Initial counselling is primarily facilitated by oncology specialists and deals with the risk of decreased fertility due to cancer treatment and, if indicated, the patient is referred to a fertility unit at the same hospital. The specialized counselling, regarding fertility preservation options and subsequent follow-up after cancer treatment, is facilitated by fertility specialists.

## 3. Results

Characteristics of the study population are presented in Table 2 below.

Three themes were identified, as presented in Table 3.

### 3.1. Support Is Needed for Navigating the Fertility Information Jungle

Most of the participants in this study indicated that initial oncofertility counselling had not been offered by the oncology specialists at diagnosis or at subsequent consultations. The AYAs independently had to request specific information about the opportunity for fertility preservation. Likewise, some participants had to ask for referral to the fertility unit to receive this information. During the cancer treatment period, very sparse fertility information was offered by the oncology specialists:

“you have learned from being through such a long process, that you have to knock on the door yourself, and phone and ask and follow up”(Participant, 35 years old)

Several participants felt somewhat responsible for their own treatment regarding fertility. This led to an overwhelming level of active decision-making. This often included dilemmas regarding choice of chemotherapy, administration of hormone substitution and finding an oocyte donor. There was no guidance from the oncology specialists. They also described having to be self-reliant and that they learned to take the initiative if they wanted oncofertility counselling:

“…as a patient you are told ‘we are in control of your chemo and we are in control of your blood tests, we are in control of everything’, but the Goserelin (a GnRH agonist causes suppression of gonadal function and is used to protect the ovaries from the gonado-toxic effects of chemotherapy), you just have to keep track of it yourself … it was almost as if my fertility was my own responsibility”(Participant, 31 years old)

Falling between two different departments and facilitating the communication on your own about fertility preservation issues were experiences shared among several participants. They felt that they were sent back and forth between different departments when, e.g., side effects of hormone replacement therapy arose. This led to the feeling of being in the middle of a negotiation about which department should be responsible for handling the given issue.

“Where there has been, as I said, a mess-up, that’s when the gynaecologist took over. She should not have done that. I should probably have just been to the fertility doctor. So, I would say the fertility doctor could probably have done the same for me, given me hormones and whatever else was needed … you are tossed back and forth between gynaecologists and the fertility clinic”(Participant, 34 years old)

Most participants felt obliged to seek initial oncofertility counselling elsewhere—and felt that the search for information was like navigating a jungle because they did not know their opportunities. In particular, the Internet, including closed Facebook groups, was used to gain information about fertility preservation, along with options regarding adoption, oocyte-donation and surrogacy. Moreover, at Kræftværket, health professionals and other AYAs were informative when the participants sought information. The participants also received information at “theme meetings” about fertility that were arranged by Kræftværket. A senior doctor from the fertility unit took part and gave information in plenary and answered questions. Some participants sought information from their relatives, who themselves had experience of oncofertility counselling or among their private networks:

“so I called an acquaintance of the family [ed. haematologist at another hospital] and said I simply need someone to tell me how to make head or tail of this, because I cannot really find out what’s going on and I am given different messages … so, yeah, I felt compelled to call another doctor because I did not trust what the oncology specialists said”(Participant, 30 years old)

### 3.2. The Doctor’s Approach Determines the Content of the Patient Consultation

During cancer treatment, most of the participants experienced that fertility was a topic the oncology specialists avoided. The participants were not asked about their thoughts or concerns about fertility, and this most obvious issue, from the participants’ perspective, was not addressed. At the first consultation with oncology specialists, some participants were given very brief information about the risks of infertility from cancer treatment, without further explanation or the chance to ask questions. Some even experienced that, when fertility was mentioned by the patient, the conversation was shut down, and there was no further dialogue on the topic:

“the same day that I get the diagnosis, the doctor also says something in the direction of ‘well, and we cannot manage to take some eggs out and that’s how it is with leukaemia, and we just have to move on from here’. So, it was a bit of an inserted sentence now, it was only because I really felt it had been shut down from day one that I did not ask more about it”(Participant, 30 years old)

Many participants described that the oncology specialists’ approach or attitude when fertility information was brought up by the patient had a significant impact on the conversation, which often closed it down. The oncology specialist’s attitude when fertility was addressed by the patient was described by several participants as embarrassed, annoyed or judgmental.

“The doctor became a little embarrassed, he did not quite know how to react”(Participant, 31 years old)

And “I feel, in a way, that my doctor gets sort of annoyed—’you cannot be a mother when you have a tumour like this in your head … it is not good for you to get pregnant because you will hurt yourself and the tumour can grow when you get pregnant’” (Participant, 20 years old).

Some participants argued that the oncology specialist’s gender at some point influenced the approach to initial oncofertility counselling. Female doctors more often seemed to understand the importance of the issue, and initiated conversations about fertility—contrary to the male doctors, who were more likely to avoid the topic.

“The doctor just said ‘my patients are usually over the age of 60 …’ you have to find someone else to ask,’ because he (the male doctor) did not know”(Participant, 31 years old)

And “Before I arrived, she (the female doctor) had actually tried to get hold of someone up at the fertility clinic so I could talk to someone even the same day, because they were probably well aware that, when you are 25, it is something that means a lot to you” (Participant, 26 years old).

The participants did not question the sincere dedication and professional skills of the oncology specialists regarding the patient’s cancer treatment. However, the risk of infertility was for several participants the most overwhelming concern at diagnosis and during treatment. Therefore, the lack of sufficient initial oncofertility counselling had a huge impact on their experience of the medical consultation during and after cancer treatment. This was especially the case when the participants tried to initiate a conversation about fertility without being met with adequate answers or an accommodating approach. One of the participants stated that the oncology specialists were pessimists, and the fertility specialists were optimists: 

“Oncologists have a very pessimistic view of it and very much a sort of dead-set view of it. They do not have that down at the fertility clinic. I also think they know how happy a baby can make a woman”(Participant, 31 years old)

In contrast to this, the participants who were referred to the fertility specialists for specialized oncofertility counselling, expressed being met with understanding, taken seriously and were reassured that the fertility specialists would help them as much as possible with fertility preservation and subsequent fertility treatment, if necessary. Moreover, the fertility specialists were up-to-date regarding the plans for the participant’s cancer treatment and knew about the risks of infertility related to the various chemotherapy regimens. This led to a great sense of relief in relation to the participants’ concerns and allowed them to focus on the cancer treatment itself and convalescence. One participant argued that, even though she unfortunately did not have any fertility preservation options due to the location of the cancer and the need for acute chemotherapy, the fact that the fertility specialists had taken the time and presented her the information as to why it was impossible, led to her being able to accept the risk of infertility:

“I felt like the opportunities I have at least heard of before, they went through them for me. It was then also explained why it was not an option, so regardless of whether you liked the answer you got or not, you were then at least clear that they had considered all the options that could have been possible”(Participant, 26 years old)

Finally, some participants mentioned that, even though they had a huge need for oncofertility counselling, it could be difficult to have an open conversation about this, especially if their relatives e.g., parents or a relatively new romantic partner participated—especially if they had never discussed this or told their parents about their family planning dreams. This was particularly difficult if the participant was unprepared for the topics of the consultation and if the risk of infertility suddenly was brought up. One participant, whose mother joined the consultations, argued that this led to a mother-daughter conflict, and the patient chose to exclude her mother from her further fertility treatment plans:

“then you have to also hear her [her mother’s] views. So, it crosses a line to have to tell about my baby plans, I also think it’s a bit annoying actually … so therefore I have not told her that I am now going for fertility treatment”(Participant, 25 years old)

### 3.3. Inadequate and Worrying Information Causes Mistrust and Frustration

Most of the participants argued that they had received an inadequate level of initial information regarding fertility from the oncology specialists and it had not been explained why the cancer treatment increased the risk of infertility. Written information regarding cancer treatment and fertility preservation options either did not exist or was too generalized. Participants with germinal cell-type ovarian cancer experienced a lack of written fertility information and received written information regarding testicular cancer and fertility preservation options, because the oncology specialists did not have specific written information for AYAs with this type of ovarian cancer. The participants did not benefit from this material—and felt insignificant as a minority group, in contrast to the larger group of elderly menopausal women who are diagnosed with ovarian cancer:

“I don’t think I’ve received much advice about it … I’ve received the same treatment as if I’d had testicular cancer, so they only had these information brochures and so on that they had for testicular cancer patients”(Participant, 25 years old)

If the participants received information, it was often too generalized and described all the options in detail. The participants argued that reading about options that were irrelevant for them, made them insecure about whether the right choice had been made for them regarding fertility preservation.

“In the written information you get, all the possibilities are given ... but of course, you don’t have all the possibilities”(Participant, 28 years old)

Several participants with cryopreserved ovarian tissue were uncertain of the circumstances surrounding the cryopreserved tissue, e.g., the location of their stored tissue and the length of time the tissue would be preserved. This led to uncertainty about the time range for fertility treatment after cancer treatment, which they didn’t feel the doctors took into account during the specialized fertility counselling. Moreover, the participants were in doubt about whether they still belonged to the fertility unit or not, and where to obtain information, since most of them only had been in contact with the fertility unit once or twice.

“The week after the operation it was like ’where is it really that my ovary is now?’ Is it frozen somewhere? It’s in a freezer ... I did not know where it was or who I should get it from if I wanted to use it”(Participant, 28 years old)

The majority of the participants had experienced receiving divergent information regarding fertility, especially from the oncology specialists. In this case, it was difficult for them to clarify which answer was the correct one. This led to a great deal of frustration and the feeling of being left alone without knowing where to turn. The divergent information concerned, e.g., how long the patient had to wait before trying to become pregnant after cancer treatment. The participants also mentioned conflicting information regarding whether the chemotherapy regime they had received would increase the risk of infertility. One participant had been through several regimes since childhood, and her mother had on numerous occasions asked the paediatric oncology specialists about the risk of infertility and been told that there was no risk. When, at age 20, the patient again had to undergo a new chemotherapy regime, she asked the oncology specialists herself about the risk of infertility. She was told that her ovaries had already been damaged due to previous chemotherapy. This shocked both the participant and her mother and led to a feeling of mistrust towards the doctors when given new information:

“The oncologist said this summer that my ovaries were already affected because I had had such strong chemo as an 8-year-old. And my mother was in shock—then she said that it couldn’t be right, because then the doctor said it was not going to affect them at all”(Participant, 20 years old)

Several participants experienced that the information they received was not only incorrect but also frightening. One participant was briefly informed about cancer treatment and the risk of infertility at her first consultation with an oncology specialist, before the patient had been informed that she had cancer:

“After surgery on the ovary, I had a conversation at hospital XX where they said it was just a normal cyst, but we’re just passing it on to Hospital YY for safety’s sake. And then I came to talk at Hospital YY, where I just thought it was like that … and then they talked to me like I knew I had cancer ... yes but it was actually cancer you had, you have to start chemo on Monday, it was sort of ‘wow’”(Participant, 25 years old)

Another participant, who had not received fertility preservation, was informed incorrectly, by an oncology specialist, that her oocytes had been genetically damaged from the chemotherapy and she would have to wait two years before trying to become pregnant. This information frightened her, because she had been reassured prior to cancer treatment that this was not the case, and now it was too late, if the new information was true. On asking other oncology specialists, no one could give her an accurate answer as to whether it was true or not.

### 3.4. Suggestions from the Participants to Improve Oncofertility Counselling

In the interviews, the participants were asked to give suggestions for possible improvements to the oncofertility counselling. The participants were free to come up with suggestions of any kind; thus, both suggestions for initial and specialized fertility counselling were proposed. The specific suggestions, along with essences from the themes regarding needs and wishes for fertility counselling, are presented in Table 4.

## 4. Discussion

In this study, we found that most participants did not receive the fertility counselling, whether initial nor specialized, that they wished. Many of the participants felt as if they were falling between departments, not knowing which department was responsible for different aspects of the fertility counselling. This testifies to a major gap between oncology and fertility specialists and that bridging this gap is essential, in order to improve AYAs’ experiences of responsibility for addressing the initial and the specialized content of oncofertility counselling [30,31]. In line with the finding in the current study, the importance of clarifying roles in a multidisciplinary team including oncology specialists, fertility specialists and other health care professionals, has been highlighted [32,33].

Several participants in our study experienced a condescending approach or response from the oncology specialists that negatively influences their relationship, which leads to dissatisfaction with the initial oncofertility counselling. Previous studies support our findings, arguing that oncology specialists’ personal assumptions can potentially have implications for the patient’s well-being and cause great concerns if the approach is considered to be patronizing [20,31,32]. Moreover, we found a difference in the approach based on the gender of the oncology specialist. Young female oncology specialists were more attentive to the importance of initial oncofertility counselling and addressed the topic more often, in contrast to male oncology specialists. This is in line with previous studies, showing that female doctors, compared to male doctors, devote more time to communicate with their patients and focus more on psychosocial and socio-emotional aspects. This leads to significantly more satisfied patients, even after adjusting for patient characteristics and physician practice style [34,35,36]. Finally, our study highlighted a need for oncology specialists to pay more attention to situational awareness, in cases where relatives attend the consultations along with the AYAs. Even though AYAs have a wish for time alone with health professionals to discuss sexual and reproductive health [37,38], doctors often have difficulty asking relatives to step outside the patient room [39]. On the other hand, it is sometimes helpful to have a friend or a relative present at consultations in which important information will be given, serving as ”another set of ears”. It could avoid confusion and help patients’ later better recall what was said or not at the visit [40].

The reason for avoidance or vague initial fertility information may be a lack of oncofertility knowledge [30,31,34] on the part of the oncology specialists or their assumptions related to the patient’s age, relationship status or prognosis [31,35,36]. Avoidance of fertility issues can also be caused by a misguided caution, if the information is discouraging, e.g., poor prognosis, or no fertility preservation options are available [23,31]. However, AYAs are dissatisfied when answers to their questions are given vaguely [41]. The results in our study show that the participants would rather know the facts instead of being spared, as they are often left with greater concerns when they are unaware of their situation and options. Due to lack of information provided, the participants in our study sought fertility information elsewhere, e.g., on the Internet. This is confirmed in several studies, also showing that, despite concerns about unreliable and confusing information, patients still use the Internet as a primary information source [40,42,43].

As regards to the content of written and spoken information handed out on the wards, our study showed that the participants often experience a lack of sufficient written and verbal fertility information. Written information is either non-existent or too generalized. The importance of accurate and individualized written information is consistent with previous studies that argue that receiving timely, detailed, and accurate spoken and written information is essential [44,45]. Moreover, some of our participants experienced completely divergent fertility information or information that appears frightening to them. In line with another study, this leads to distress and confusion, negatively influencing the patient’s relationship with and trust in the knowledge of the oncology specialist [42].

On the positive side, in our study we found that those few participants who received specialized fertility counselling in addition to initial fertility information from the oncology specialists expressed greater satisfaction and comfort. When specialized fertility information was given, the participants were more likely to accept their current situation and felt more satisfied with the oncofertility counselling compared to those who received inadequate fertility information. Moreover, a focus on post-cancer treatment follow-up with fertility assessment and sexual and reproductive health counselling was more likely to be conducted when the AYAs were referred to fertility specialists. This had a positive impact on their experience of oncofertility counselling. The need for greater access to fertility clinics is described in the literature [40,44,46,47].

The suggestions for fertility counselling proposed by the participants in the current study are consistent with recommendations and studies in the field, e.g., regarding doctors’ communication skills and psychosocial and family support [4,40,44]. Attention should be paid to youth-friendly communication when engaging with AYAs, including how to connect, being watchful and attentive to AYAs’ needs, and being respectful, supportive, and caring [48,49,50]. Greater satisfaction is reported from cancer patients who receive fertility information in an honest and respectful manner [33]. The suggestions for fertility counselling from our participants may be a useful tool for doctors offering fertility counselling. However, the statements in Table 3 represent solely the viewpoint of the interviewed patients and do not reflect the opinion of any of the authors. Future studies should investigate whether these suggestions can and should be implemented.

### Strengths and Limitations

A strength in this study is the use of qualitative interviews, which allowed for an in-depth examination of the participants’ experiences of oncofertility counselling. Moreover, the study participants represented a wide variation in demographics, including a balanced age distribution and relationship status—including single AYAs and those in relationships. The population represented five of the most common cancers in AYAs in Denmark [51]. Fifty percent of the participants in our study had a master’s degree, and the rest had a bachelor’s degree or were currently undergraduates. This indicates that our study population is not representative of all female AYAs, especially those with lower socioeconomic status. This supports the importance of initiating oncofertility counselling, to prevent inequality in health and marginalization.

The study was limited by being conducted at a single centre, limiting general statements on how widespread in Denmark is the lack of initial and specialized oncofertility counselling. However, international studies have also addressed this issue [40,52].

The study was undertaken at a regular hospital ward transferable to other national and international hospital settings with the same patient population, which strengthen the external validity of the study [27].

With the international definition of AYAs being age 15–39 [53] a possible limitation in this study is the exclusion of AYAs age 15–17 and that the youngest AYAs participating in this study being 20 years old. However, the lower limit of 18 years was deliberately chosen because young female AYAs at Kræftværket has expressed a need for oncofertility counseling without including parents due to the sensitivity of the topic. In Denmark the legal age is 18, why we were unable to include patients younger than 18 without an informed consent from their parents. We acknowledge the importance of examining the experience of oncofertility counselling towards AYAs with cancer from age 15, hoping to conduct a study with this focus in the future. Finally, there was a possible selection bias in that patients with a special interest in fertility issues might have been more eager to sign up to interviews.

## 5. Conclusions

Results in this single centre, qualitative study indicate a continuing problem regarding insufficient oncofertility counselling to AYAs with cancer. In our setting, this implies the need for further improvement to ensure uniform and adequate information, especially in the initial oncofertility counselling. National clinical guidelines for oncology specialists are suggested, including information about cancer treatment and fertility, fertility preservation options and characteristics of patients who should be referred to fertility specialists. We believe, it is necessary to be aware of the importance of the multidisciplinary effort and the collaboration between oncology and fertility specialists at time of diagnosis, during cancer treatment and post-treatment. In addition, participants in this study, suggest focus on verbal and written information along with communication upskilling to improve oncofertility counselling.

## Figures and Tables

**Table 1 cancers-13-01355-t001:** Sample questions from the semi structured interview guide.

Thoughts of Family and Fertility	What Were Your Thoughts on Family Planning/Having More Children Prior to the Cancer? What Were Your Thoughts about Fertility at Diagnosis and during Cancer Treatment? What Are Your Thoughts Now Regarding the possibility of Having Children in the Future?
The fertility counselling	Describe the fertility counselling you received at diagnosis and during cancer treatment. What verbal and/or written information about fertility did you receive? Who initiated the topic fertility at consultation? Did you seek information about fertility and cancer treatment elsewhere—if yes, where? Were you referred to a fertility specialist for specialized fertility counselling? If yes, did you have to request this yourself?
Experience of counselling	What was your experience of the fertility counselling? Did you feel sufficiently informed about fertility and your fertility options?—if not, what was missing?
Suggestions	Do you have suggestions how to improve the oncofertility counselling for AYAs in the future?

**Table 2 cancers-13-01355-t002:** Demographic variables.

		Participants *n* = 12
Mean age (range)		28 (20–35)
In Treatment		4
Post treatment		8
Relationship status		
	In a relationship	8
	Single	4
Education		
	High school	4
	Bachelor’s degree	2
	Master’s degree	6
Cancer type		
	Acute Myeloid Leukaemia (AML)	1
	Acute Lymphoblastic Leukaemia (ALL)	1
	Brain cancer	1
	Breast cancer	2
	Hodgkin’s Lymphoma	3
	Non-Hodgkin’s Lymphoma	1
	Ovarian cancer	3

**Table 3 cancers-13-01355-t003:** Results from the qualitative interview divided into three themes.

Theme	Summary of Contents
Support is needed for navigating the fertility information jungle	Oncofertility counselling request Responsibility for own treatment Falling between departments Seeking information elsewhere
The doctor’s approach determines the content of the patient consultation	Fertility was a topic that the oncology specialists avoided Oncology specialists’ approach and attitude when fertility was addressed Female versus male doctors Oncology specialists versus fertility specialists Situational awareness when relatives participates in the consultation
Inadequate and worrying information causes mistrust and frustration	Non-existent or too generalized information Uncertainty about the circumstances surrounding the cryopreserved tissue Divergent and frightening information led to confusion and mistrust

**Table 4 cancers-13-01355-t004:** Suggestions from the participants on how to improve the oncofertility counselling.

**Verbal Information**
Mandatory information: Initial oncofertility counselling should be mandatory when AYAs are informed about the diagnosis Chance of pregnancy: Inform about the likelihood of becoming pregnant after cancer treatment Contraception: Inform about contraception during cancer treatment Side effects: Inform about the risk of climacteric symptoms and the possibility of hormone substitution If there are questions: Inform the patient where to call if they have questions Private oocyte donation: Mention Facebook groups regarding private oocyte donation
**Written information**
All diagnoses: Written information about fertility, cancer treatment and fertility preservation options for every cancer diagnosis Upon diagnosis: Hand out the written information upon diagnosis Alignment: Verbal and written information must be adapted and adjusted to the individual cancer patient Accessibility: Written information must be accessible—e.g., webpage, app, brochure Options and hope: Mention briefly fertility treatment including oocyte donation and success stories
**Timing of information during cancer treatment**
Initiative: Oncofertility counselling should be initiated by the oncology specialists Readiness: Inform that fertility can be discussed when the patient is ready—also later during the cancer treatment period Individually adapted: Oncofertility counselling must be individually adapted Continuity: If possible, let the patient have the same team of oncology specialists during the time of diagnosis and cancer treatment Alignment of expectations: Present the topic “fertility” when arranging the next consultation, so the patient can bring the most suitable relative Follow up: Arrange one or more follow-up consultations regarding the patients’ concerns about fertility (or other topics) Genetic test: When applicable, offer a rapid DNA test–it is significant in relation to fertility preservation options
**Co-operation between specialists**
Referral: Referral to the fertility specialists by oncology specialists at diagnosis Alignment: Oncology specialists and fertility specialists must coordinate the information the patients are given Contact person: Arrange a contact person, so the patient knows who to contact if they have questions
**Specialist’s communication skills**
Honesty: Be honest about the risks of infertility Instil hope: Keep in mind that the AYAs with cancer still have hopes and dreams for the future after cancer treatment Awareness: Regardless of gender, age and medical specialism, the doctor must be aware of the importance of fertility counselling Equality: Meet the patients at “eye level”, addressing and respecting the topics important to the patient Positive attitude: Be open-minded towards the patients with hopes for the future
**Fertility unit and preservation options**
Fertility preservation: Inform the patient where and for how long the cryopreserved tissue/oocytes/embryos/sperm is preserved Routine procedure: Cryopreservation of ovarian tissue/oocytes/embryos and sperm deposits ^1^ should be a routine procedure when AYAs are diagnosed with cancer and in risk of infertility Fertility examinations: Offer the possibility of fertility assessment after cancer treatment Partner only: Offer a consultation for the partner only, at the fertility unit

^1^ Stated by patients with a male partner.

## Data Availability

All data associated with this study is present in this paper.

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
