# Peer review of "How Do Young Women with Cancer Experience Oncofertility Counselling during Cancer Treatment? A Qualitative, Single Centre Study at a Danish Tertiary Hospital"

_cancers, 2021, doi:10.3390/cancers13061355_

Round 1

Reviewer 1 Report

It was a pleasure to read such a well-written and highly significant piece of work.

I have just some minor comments which might improve the manuscript. In the introduction, take care not to repeatedly mention that the number of AYAs receiving fertility information / counselling is low.

(p.3) The introduction could also merit from an overview of the centre / group (Kræftværket) from where the participants were recruited.

Please outline a rationale for your definition of AYAs. I note that the eligibility criteria for the study was 18 years and above wand that your youngest participant was aged 20 years. It would have been interesting to capture the experiences of younger AYAs (e.g., 13 years and above).

Method

In terms of sample size, I recommend you stating that recruitment will continue until data saturation and define what you mean by this. The sample size of 12 is somewhat small and given that recruitment took place from one centre and that the sample was likely to be more homogeneous in their opinions, there will be limitations in terms of generalisability which I appreciate you acknowledge in the discussion. Extending recruitment across different centres might have resulted in more diverse experiences.

Please include the interview schedule as supplementary material or provide more detail regarding the questions.

Results

Please present the characteristics of the sample in the results rather than the methods.

Author Response

Dear Reviewer 1

Please see the attachment including a cover letter to the Editor and all three Reviewers at Cancers and answers to the rewview comments. 

Reviewer 2 Report

This is a well designed single institution study addressing a much needed knowledge gap in AYA oncofertility research. Simple Summary: Pg 1 Ln 17/18: consider changing language from "of pregnancy" to "for pregnancy" and also switching the word "ending" to "completion." Pg 1 Ln 21-23: please clarify if this study has one aim with the sub-aim of increasing awareness of two aims and the second aim is increasing awareness. Consider omitting the pat of the sentence starting with "e.g...." For a summary of the paper, it is not clear what this means. Abstract: Pg 1 Ln 24: the word "cancer" can be removed from "cancer therapy-induced fertility." This is understood as earlier in the sentence its noted that this is about AYAs with cancer. Pg 1 Ln 33-35: as the themes currently read, its hard to tease out exactly what they mean or represent so pls consider adding a little additional text to each theme to help them have meaning when they stand along like this in the abstract. Pg 1 Ln 38: I'm not familiar with the word "upskilling" - maybe consider another word here. Introduction: Pg 2 Ln 64: is this meant to start a new paragraph, if so the indentation is missing. Methods: Pg 3 Ln 97: can you provide a bit more text (not much is needed) on what a "social organization" means in this specific case? Pg 3 Ln102/103: can you provide a little bit more information about how many patients are a part of the closed facebook group or how many actively post/use this group. That will help give the reader a sense of how generalizable the study sample is for the group represented through this Facebook group. Pg 3 Ln 115/116: Also, if possible, could you please provide some evidence to support that a sample size of 10-12 is an acceptable number to demonstrate thematic saturation? Pg 4 Ln 147: "Definition" should be changed to pleural "Definitions." Results: Table 2: as noted above, it might be helpful to the reader reviewing the abstract or Table 2 only to have themes that have slightly more text and communicate a free-standing message when read in isolation. For example, for the first theme on navigation; based on the summary of contents in Table 2, it could read "Support is needed for navigating the fertility information jungle." That is just an example but it could help clarify the themes to readers. Table 3: this table is hard to read/follow to this reviewer, could it be aligned on the left instead of centered. that might be all that is needed to make it more readable. Could an additional table be added that delineates participant number and then lists their diagnosis and age? This would give more meaning to the inclusion of each participant number after quotes. Discussion: Pg 9 Ln 365: Please clarify what is meant by "'...experiences of role division.." Pg 9 Ln 373: what personal assumptions do you reference here? Pg 9 Ln 381: Is "addressed" the correct word here? Maybe it "highlighted" a need... Pg9 Ln 385: consider adding the words "...the patient room." at the end of the sentence. Pg 10 Ln 387: I'm not sure "memorizing" is what the authors meant here...maybe an alternate way of saying this is "...help patients later better recall what was said or not at the visit." Conclusions: I would just urge the authors to be very deliberate and careful with language here in this section and not make too many generalizing statements about the findings of this study as this reflects a very limited sample (both in terms of size and that its a single institution study). These conclusions may very well be complete true for the 12 participants interviewed or potentially this one center's AYA cancer population but beyond that, it will be difficult to extend these findings beyond this setting, care environment, institutional model of care, etc.

Author Response

Dear Reviewer 2

Please see the attachment including a cover letter to the Editor and all three Reviewers at Cancers and answers to the review comments. 

Reviewer 3 Report

Dear Author,

I read with great interest the manuscript titled “How do young women with cancer experience oncofertility counselling during cancer treatment? A qualitative, single centre study at a Danish tertiary hospital”.

Authors performed a qualitative study on 12 young female adults with cancer or cancer survivors asked to participate at a semi-structured interviews regarding oncofertility counselling

General considerations:

The study deals with a very important topic already stated in the the Clinical Practice guidelines of the most important international society of oncology

in the ASCO Clinical Practice Guideline on fertility preservation in patients with cancer, updated in 2018:

“Health care providers should initiate the discussion on the possibility of infertility with patients with cancer treated during their reproductive years or with parents/guardians of children as early as possible. Providers should be prepared to discuss fertility preservation options and/or to refer all potential patients to appropriate reproductive specialists. Although patients may be focused initially on their cancer diagnosis, providers should advise patients regarding potential threats to

fertility as early as possible in the treatment process so as to allow for the widest array of options for fertility preservation. The discussion should be documented”

and in ESMO Clinical Practice Guidelines on fertility preservation and post-treatment pregnancies in post-pubertal cancer patients, published in 2020:

“All cancer patients of reproductive age should receive complete oncofertility counselling as early as possible in the treatment planning process, irrespective of type and stage of disease. This should include discussion of the patients’ current or future family desire, their health and prognosis, the potential impact of the disease and/or proposed anticancer treatment on their fertility and gonadal function, chances of future conception, pregnancy outcomes and offspring, as well as the need for effective contraception in the context of systemic anticancer treatment…”

Although the topic is interesting, and the study is well conducted, It seems more suitable as an internal investigation rather than the public interest, moreover the results are also quite critical.

Specific considerations:

Even if data saturation was achieved after the first 10 interviews, twelve patients are quite small a sample to draw conclusions: the sample is very heterogeneous both by type of tumor and by age of the patients. It is not known whether the patients already had children or not, or whether they had or not desire of pregnancy (as you stated in lines 436-438)

The results are presented rather vaguely: "most of the participants" (line 164); "some participants" (line 167); "several participants" (line 172); "most participants" (line 193) etc ...

I suggest not to use drug brand names, even in patient reported comments (Line 179: Zoladex)

Table number 3 shows the suggestions from the participants on how to improve the oncofetility counseling. This table shows the need for information on sperm preservation ... but the participants are all female and sperm preservation is not applicable

In the discussion section Autors stated: “we found a difference in the approach based on the gender of the oncology specialist. Young female oncology specialists were more attentive to the importance of initial oncofertility counselling and addressed the topic more often, in contrast to male oncology specialists” but I have found no data in the materials and methods section nor in the results as to whether the doctors were male or female, young or old.

Author Response

Dear Reviewer 3

Please see the attachment including a cover letter to the Editor and all three Reviewers at Cancers and answers to the review comments. 

Round 2

Reviewer 2 Report

Thank you for addressing the changes previously suggested.

Author Response

Dear reviewer 2

Thank you for your commments - they have, in our opinion, improved the manuscript.

Best wishes,

Line Bentsen